# Secure Transmission for Simultaneous Wireless Information and Power Transfer in AF Untrusted Relay Networks

**DOI:** 10.3390/s19010076

**Published:** 2018-12-26

**Authors:** Hui Shi, Weiwei Yang, Dechuan Chen, Yunpeng Luo, Yueming Cai

**Affiliations:** 1Army Engineering University of PLA, No. 88 Houbiaoying, Qinhuai District, Nanjing 210007, China; lgdhpkry@aliyun.com (H.S.); luoyunp@aliyun.com (Y.L.); caiym@vip.sina.com (Y.C.); 2Wuhan Zhong Yuan Electronics Group Company Ltd., Wuhan 430205, China; Chenchuan927@163.com

**Keywords:** energy harvesting, amplify-and-forward, multiple antennas, untrusted relay, outage probability, effective secrecy throughput

## Abstract

This paper investigates secure communications of energy harvesting untrusted relay networks, where the destination assists jamming signal to prevent the untrusted relay from eavesdropping and to improve the forwarding ability of the energy constrained relay. Firstly, the source and the destination transmit the signals to the relay with maximal ratio transmission (MRT) technique or transmit antenna selection (TAS) technique. Then, the destination utilizes maximal ratio combining (MRC) technique or receive antenna selection (RAS) technique to receive the forwarded information. Therefore, four transmission and reception schemes are considered. For each scheme, the closed-form expressions of the secrecy outage probability (SOP) and the connection outage probability (COP) are derived. Besides, the effective secrecy throughput (EST) metric is analyzed to achieve a good tradeoff between security and reliability. In addition, the asymptotic performance of EST is also considered at the high signal-to-noise ratio (SNR). Finally, simulation results illustrate that: (1) the EST of the system with MRT and MRC scheme are superior to other schemes, however, in the high SNR regime, the EST of the system with MRT scheme is inferior to TAS; and (2) for the source node, there exists an optimal number of antennas to maximize the EST of the proposed schemes.

## 1. Introduction

Because of the inherent broadcasting characteristics of the wireless channel, wireless communications are vulnerable to be attacked by the eavesdroppers through the public channel. The traditional method to protect the information privacy is the assistance of upper-layer cryptographic secure schemes, which mainly depends on computation complexity. However, with the rapid advancement of the computing capability, the security of the wireless communication encounters tremendous challenges and risks. In view of this, the physical layer security (PLS) has been proposed as a complement of current cryptographic method, which takes advantage of time-varying characteristics and the randomness of the wireless channel. The research on PLS was pioneered by Wyner, who introduced the degraded wire-tap channel and gave the definition of secrecy capacity [1]. The notion of secrecy capacity measures the maximal secrecy rate at which the confidential information can be transmitted to the destination secretly and reliably [1,2,3].

Recently, cooperative communication is an effective way to extend the coverage of the transmission and to increase the secrecy capacity. The performance of the dual-hop relay network has been studied for both the single relay and multiple relays under Rayleigh fading channels in [4,5,6,7]. In [6], the authors significantly improves the security of the system with source relay selection scheme. Taking the place of full-duplex relaying, two-path successive relaying protocol is proposed to reduce the probability of being intercepted [7]. However, the relay may not be trusted, which is also a potential eavesdropper [8,9,10,11,12,13,14,15,16]. Several works about the untrusted relay has been published, where the destination node transmits the jamming signals to the untrusted relay for securing the transmissions [9,10,11,12,13,14,15]. In [16], a destination-assisted jamming scheme is put forward and the instantaneous secrecy rate is studied in multiple amplify-and-forward (AF) untrusted relay networks. In [17,18], a jamming signal is designed in a smart way. The artificial noise symbols are designed and transmitted to degrade the error probability performance [17]. The artificial noise parameter is derived for two kinds of optimal points, max-min point and single user point [18].

Ref. [19] considers an untrusted relay cognitive Internet of Things (IoT) network, where the probability of successfully secure transmission metric is investigated to evaluate the security of the network. To further improve the secrecy performance of cooperative networks, multiple antennas technique has been implemented into the system [20,21,22]. Both the source and the destination are multi-antennas, while the relay is only one antenna in the considered network [20]. Ref. [21] studies the AF multiple-input multiple-output (MIMO) system with untrusted relay; besides, both the destination and the relay are multiple antennas. In [22], a large-scale MIMO relaying system with an energy harvesting relay is investigated, and the secrecy performance of the system under imperfect channel state information (CSI) is analyzed.

Utilizing the precoding techniques could also improve the secrecy performance of the cooperative networks [23,24,25,26]. In [23], precoding at both the source and the relay scheme is proposed. To maximize the secrecy rate, the source, relay, and destination precoding matrices are jointly designed. In [24], to maximize the secrecy sum rate, the source’s signal, artificial noise precoders and the relay’s precoder, under the sources and relay transmit power constraints, are jointly designed. In [25], the authors investigates the source and relay precoders that characterizes the relationship between the achievable stored energy at the relay and the achievable source-to-destination rate. In [26], the authors examines the maximal energy efficiency by design of the optimum precoding matrices at the transceivers and the relay node as well as the optimum power splitting factor for SWIPT.

In some special application scenarios, many energy-constrained cooperative relays are configured to cover larger districts where recharging and exchanging relay batteries are not convenient and economical [27]. Energy harvesting (EH) has been developed as a promising energy source for dealing with this problem [28,29,30]. Moreover, EH and information processing simultaneously has been applied in cooperative relaying [31,32,33,34,35], where they relay harvests energy from the receipt signals and then forwards the information to the destination via the received power. To harvest energy and decode information from the received signals, the power splitting relaying (PSR) and the time switching relaying (TSR) are proposed [36,37,38]. Besides, EH is considered in energy-free Internet of Things (IoT) device within the bistatic scatter (BS) mode [39].

The effects of EH relay on the secrecy performance has been studied in [38,40,41]. In [38], the authors focuses on an AF two-way relay network employing the multiple access broadcast protocol and TSR protocol, where the legitimate communication parties are multi-antennas while the assisted relay is only one antenna. However, the communication parties adopt MRT/MRC techniques. Moreover, the relay is considered to be trusted, thus only the connection outage probability (COP) is considered for MRT/MRC techniques, and the secrecy outage probability (SOP) performance is not analyzed. The SOP and COP metrics of the system is analyzed in [40]. The SOP performance is analyzed over Nakagami-m Fading Channels [41,42]. The authors improves the harvested energy with multiple antennas, and studies the maximal harvested power by jointly different parameters [43,44,45]. The cooperative jamming is considered in [45], and the maximal secrecy rate is investigated. The author studies the secrecy transmission with the unreliable wireless backhaul link, and derives the closed-expressions for SOP ergodic secrecy rate and non-zero achievable secrecy rate to examine the effect of the backhaul reliability on the system performance [46]. However, each node is equipped with a single antenna and all relays are considered to be trusted.

Some efforts on the multi-user scenarios have been considered in [47,48,49,50]. In [47], the authors investigates the relay network consisting of a source, a set of relays and users in the presence of an eavesdropper. Each terminal is equipped with a single antenna. The cooperative beamforming and user selection techniques are applied to protect the confidential message. The secrecy rate (SR) and SOP of the network are investigated. Dong-Hua Chen et al. [48] investigates full-duplex (FD) secure transmissions with simultaneous wireless information and power transfer (SWIPT) on downlink (DL), where an FD base station, with multiple transmit antennas and multiple receive antennas, serves a group of single-antenna users on both DL and uplink (UL). The minimization of the system power under constraints on EH and security rates is examined. In [49], a MIMO relay system consists of one base station, one relay station, multiple legitimate users and malicious eavesdroppers. The system worst-case secrecy rate is maximized by jointly optimizing base station beamforming matrix, the relay station amplify-and-forward transformation matrix, and the covariance matrix of artificial noise. In [50], the authors investigates the secrecy performance of multiuser dual-hop relay networks where a base station communicates with multiple legitimate users through the assistance of a trustful regenerative relay in the presence of multiple eavesdroppers. To evaluate the secrecy performance of the considered system, two practical situations are addressed based on the availability of eavesdropper’s CSI, where the eavesdropper’s CSI is unavailable or available at the relay. The new exact and asymptotic closed form expressions for the SOP and the ergodic secrecy rate (ESR) are accordingly derived. However, the relay in the aforementioned multiple users system model is trusted.

Thus, few works have examined the SOP performance of the SWIPT untrusted relay MIMO system. Besides, the trade-off between the SOP and COP has not been studied in the above literature. Note that the SOP and COP are used for characterizing the security and reliability performance of the system, respectively. However, the SOP and COP are a pair of contradictory metrics. Thus, the trade-off between SOP and COP also deserves careful consideration. Motivated by the above, this paper concentrates on an AF untrusted relay network, where both the source and the destination are multi-antennas. To reduce the computational complexity, the relay is equipped with a single antenna. Hence, to enhance the PLS of the system, four transmission and reception schemes are considered, i.e., transmit antenna selection-receive antenna selection (TAS-RAS), maximal ratio transmission-receive antenna selection (MRT-RAS), transmit antenna selection-maximal ratio combining (TAS-MRC), and maximal ratio transmission-maximal ratio combining (MRT-MRC). The main contributions of this work can be summarized as follows.

The SOP and COP performance of the energy harvesting multi-antenna untrusted relay network under PSR are studied. The paper further researches on the correlation between SOP and COP using the metric of effective secrecy throughput (EST). The SOP, COP and EST expressions of closed form are orderly derived for the proposed schemes, which provide deep insights into the effect of the system performance. In addition, the signal-to-noise ratio (SNR), the antennas number at the source and other parameters were also researched.When the transmit SNR goes to infinity, a comprehensive study on the asymptotic analysis of the EST was conducted to elicit the optimal energy harvesting strategy for the network. The deduced expressions and simulation results illustrate that, for the four schemes, the EST approaches to a constant at high transmit SNR. The EST of the TAS-MRC scheme is superior to the three other schemes. In addition, the effect of the power allocation coefficient, the power splitting ratio factor and the number of antennas were researched on the EST.

The remainder of the paper is organized as follows. Section 2 introduces the system model and describes TAS-RAS scheme, MRT-RAS scheme, TAS-MRC scheme and MRT-MRC scheme, respectively. Section 3 presents the SOP, COP and EST expressions of the four schemes. Besides, the asymptotic expressions of the EST at high transmit SNR are investigated. Then, simulation analyses are discussed in Section 4. Finally, Section 5 summarizes the whole paper.

## 2. System Model

Consider a SWIPT untrusted relay MIMO system as shown in Figure 1, which consists of a source *S* and a destination *D*, equipped with $N_{s}$ and $N_{d}$ antennas, respectively. An untrusted energy harvesting relay *R* is equipped with a single antenna. It is assumed that the perfect CSI of all channels are available [51]. Each node operates in half-duplex mode and the AF protocol is adopted. $h_{\mathrm{SR}}$ denotes the $N_{s}\times 1$ channels vector between *S* and *R* links, and the $h_{\mathrm{RD}}$ stands for the $1\times N_{d}$ channels vector between *R* and *D* links. $h_{SR}^{*}$ and $h_{RD}^{*}$ denote channel coefficients between the source and the relay, between the relay and the destination, respectively. ${\left|h_{SR}^{*}\right|}^{2}$ and ${\left|h_{RD}^{*}\right|}^{2}$ represent the maximal channel gain selected from the multiple antennas, i.e., ${\left|h_{SR}^{*}\right|}^{2}=\arg\max_{1\leq s\leq N_{s}}{\left|h_{SR}^{s}\right|}^{2}$ and ${\left|h_{RD}^{*}\right|}^{2}=\arg\max_{1\leq d\leq N_{d}}{\left|h_{RD}^{d}\right|}^{2}$. Then, ${\left|h_{SR}^{*}\right|}^{2}$ and ${\left|h_{RD}^{*}\right|}^{2}$ follow exponential distribution with means ${\bar{\gamma }}_{SR}$ and ${\bar{\gamma }}_{RD}$,respectively. It is assumed that each channel is independent and identically distributed (i.i.d.) and undergoes Rayleigh fading. In addition, the channels between the relay and destination are reciprocal, i.e., $h_{RD}^{*}=h_{DR}^{*}$ [15,16]. For simplicity, all noises are additive white Gaussian noise (AWGN) with zero mean and variance $N_{0}$.

For the PSR, the entire communication is divided into two phases. In the first phase, the source transmits the confidential information to the untrusted relay. To enhance PLS, the destination simultaneously sends the jamming signal to prevent the untrusted relay from eavesdropping the confidential information. The relay harvests energy from the information sent by the source and the jamming signal transmitted by the destination. In the second phase, the relay employs the harvested energy to forward the confidential information. The jamming signal can be eliminated via the self-interference cancellation techniques [52]. It is assumed that the harvested energy at *R* is divided into two proportions. The ω denotes the allocation coefficient of the harvested energy, $\omega \in \left(0,1\right)$. The ω proportion of the harvested energy is utilized for forwarding the confidential information to *D* and the other proportion is used for eavesdropping and decoding consumption.

Based on different proceed strategy at *S* and *D*, the transmission and reception of four schemes in the above system are considered.

### 2.1. TAS-RAS Scheme

For the TAS-RAS scheme, during the first phase, *S* selects only a single transmit antenna from $N_{s}$ antennas to send the confidential signal with power βP, where *P* represents the total transmitted power of the *S* and *D*, and β denotes the power allocation coefficient between *S* and *D*, where 0<β≤1. Concurrently, *D* selects a single transmit antenna from $N_{d}$ antennas to transmit the jamming signal with allocated power $\left(1-\beta \right)P$. The signal power received by *R* is split into two parts. The ρ portion of the received power is used for harvesting energy, while the rest power is applied for transmitting information to *D*, where 0<ρ<1. Hence, the received instantaneous signal-to-interference-plus-noise rate (SINR) at *R* can be written as
(1)$$\begin{array}{l} \gamma_{R}^{TAS-RAS}=\frac{\left(1-\rho \right)\beta \lambda {\left|h_{SR}^{*}\right|}^{2}}{\left(1-\rho \right)\left(1-\beta \right)\lambda {\left|h_{RD}^{*}\right|}^{2}+1}, \end{array}$$
where $\lambda =P/N_{0}$ denotes the transmit SNR.

From Equation (Equation 1), the transmit power at *R*, $P_{R}^{TAS-RAS}$, is given by
(2)$$\begin{array}{l} P_{R}^{TAS-RAS}=\omega \eta \rho \left(\beta P{\left|h_{SR}^{*}\right|}^{2}+\left(1-\beta \right)P{\left|h_{RD}^{*}\right|}^{2}\right), \end{array}$$
where η denotes the energy conversion efficiency factor of the harvested energy, 0<η<1, and ω denotes the allocation coefficient of the harvested energy; the ω proportion of the harvested energy is utilized for forwarding the confidential information to *D*.

In the second phase, *R* forwards the information to *D* with the power normalization coefficient $1/\sqrt{\beta P{\left|h_{SR}^{*}\right|}^{2}+\left(1-\beta \right)P{\left|h_{RD}^{*}\right|}^{2}+N_{0}}$. Therefore, the received instantaneous SINR of *D* can be calculated as
(3)$$\begin{array}{l} \gamma_{D}^{TAS-RAS}=\frac{\omega \eta \rho \left(1-\rho \right)\beta \lambda^{2}{\left|h_{SR}^{*}\right|}^{2}{\left|h_{RD}^{*}\right|}^{2}\left(\beta {\left|h_{SR}^{*}\right|}^{2}+\left(1-\beta \right){\left|h_{RD}^{*}\right|}^{2}\right)}{\lambda \left(\omega \eta \rho {\left|h_{RD}^{*}\right|}^{2}+1-\rho \right)\left(\beta {\left|h_{SR}^{*}\right|}^{2}+\left(1-\beta \right){\left|h_{RD}^{*}\right|}^{2}\right)+1}. \end{array}$$

### 2.2. MRT-RAS Scheme

For the MRT-RAS scheme, *S* firstly transmits the confidential information to *R* with $N_{s}$ antennas to maximize SNR of the relay. At the same time, *D* selects a single transmit antenna from $N_{d}$ antennas to transmit the jamming signal. Thus, the received instantaneous SINR of *R* can be written as
(4)$$\begin{array}{l} \gamma_{R}^{MRT-RAS}=\frac{\left(1-\rho \right)\beta \lambda {∥h_{\mathrm{SR}}∥}^{2}}{\left(1-\rho \right)\left(1-\beta \right)\lambda {\left|h_{RD}^{*}\right|}^{2}+1}, \end{array}$$
where ${∥•∥}^{2}$ denotes the Frobenius norm.

The transmit power at *R* is denoted by $P_{R}^{MRT-RAS}$, which is calculated as
(5)$$\begin{array}{l} P_{R}^{MRT-RAS}=\omega \eta \rho \left(\beta P{∥h_{\mathrm{SR}}∥}^{2}+\left(1-\beta \right)P{\left|h_{RD}^{*}\right|}^{2}\right). \end{array}$$

The *R* forwards the information to *D* with the power constraint factor $1/\sqrt{\beta P{∥h_{\mathrm{SR}}∥}^{2}+\left(1-\beta \right)P{\left|h_{RD}^{*}\right|}^{2}+N_{0}}$. Thus, the received instantaneous SINR of *D* can be written as
(6)$$\begin{array}{l} \gamma_{D}^{MRT-RAS}=\frac{\omega \eta \rho \left(1-\rho \right)\beta \lambda^{2}{∥h_{\mathrm{SR}}∥}^{2}{\left|h_{RD}^{*}\right|}^{2}\left(\beta {∥h_{\mathrm{SR}}∥}^{2}+\left(1-\beta \right){\left|h_{RD}^{*}\right|}^{2}\right)}{\lambda \left(\omega \eta \rho {\left|h_{RD}^{*}\right|}^{2}+1-\rho \right)\left(\beta {∥h_{\mathrm{SR}}∥}^{2}+\left(1-\beta \right){\left|h_{RD}^{*}\right|}^{2}\right)+1}. \end{array}$$

### 2.3. TAS-MRC Scheme

In the TAS-MRC scheme, in the first phase, *S* transmits the information to *R* with only a single antenna. Meanwhile, *D* sends the jamming signal to *R* with $N_{d}$ antennas. Hence, the received instantaneous SINR of *R* is given by
(7)$$\begin{array}{l} \gamma_{R}^{TAS-MRC}=\frac{\left(1-\rho \right)\beta \lambda {\left|h_{SR}^{*}\right|}^{2}}{\left(1-\rho \right)\left(1-\beta \right)\lambda {∥h_{\mathrm{RD}}∥}^{2}+1}. \end{array}$$

The transmit power at *R* is denoted by $P_{R}^{TAS-MRC}$, which is written as
(8)$$\begin{array}{l} P_{R}^{TAS-MRC}=\omega \eta \rho \left(\beta P{\left|h_{SR}^{*}\right|}^{2}+\left(1-\beta \right)P{∥h_{\mathrm{RD}}∥}^{2}\right). \end{array}$$

During the second phase, *R* forwards the received information with the power normalization coefficient $1/\sqrt{\beta P{\left|h_{SR}^{*}\right|}^{2}+\left(1-\beta \right)P{∥h_{\mathrm{RD}}∥}^{2}+N_{0}}$, and *D* adopts MRC technique to receive the information. Hence, the received instantaneous SINR of *D* can be calculated as
(9)$$\begin{array}{l} \gamma_{D}^{TAS-MRC}=\frac{\omega \eta \rho \left(1-\rho \right)\beta \lambda^{2}{\left|h_{SR}^{*}\right|}^{2}{∥h_{\mathrm{RD}}∥}^{2}\left(\beta {\left|h_{SR}^{*}\right|}^{2}+\left(1-\beta \right){∥h_{\mathrm{RD}}∥}^{2}\right)}{\lambda \left(\omega \eta \rho {∥h_{\mathrm{RD}}∥}^{2}+1-\rho \right)\left(\beta {\left|h_{SR}^{*}\right|}^{2}+\left(1-\beta \right){∥h_{\mathrm{RD}}∥}^{2}\right)+1}. \end{array}$$

### 2.4. MRT-MRC Scheme

For the MRT-MRC scheme, *R* firstly receives the information from *S* with $N_{s}$ antennas and the jamming signal from *D* with $N_{d}$ antennas, respectively. Hence, the received SINR of *R* can be given by
(10)$$\begin{array}{l} \gamma_{R}^{MRT-MRC}=\frac{\left(1-\rho \right)\beta \lambda {∥h_{\mathrm{SR}}∥}^{2}}{\left(1-\rho \right)\left(1-\beta \right)\lambda {∥h_{\mathrm{RD}}∥}^{2}+1}. \end{array}$$

The transmit power of *R* is denoted by $P_{R}^{MRT-MRC}$, which can be expressed as
(11)$$\begin{array}{l} P_{R}^{MRT-MRC}=\omega \eta \rho \left(\beta P{∥h_{\mathrm{SR}}∥}^{2}+\left(1-\beta \right)P{∥h_{\mathrm{RD}}∥}^{2}\right). \end{array}$$

In the second phase, *R* forwards the information with the power constraint factor $1/\sqrt{\beta P{∥h_{\mathrm{SR}}∥}^{2}+\left(1-\beta \right)P{∥h_{\mathrm{RD}}∥}^{2}+N_{0}}$, and *D* applies MRC technique to receive the information. Therefore, the received SINR of *D* can be calculated as
(12)$$\begin{array}{l} \gamma_{D}^{MRT-MRC}=\frac{\omega \eta \rho \left(1-\rho \right)\beta \lambda^{2}{∥h_{\mathrm{SR}}∥}^{2}{∥h_{\mathrm{RD}}∥}^{2}\left(\beta {∥h_{\mathrm{SR}}∥}^{2}+\left(1-\beta \right){∥h_{\mathrm{RD}}∥}^{2}\right)}{\lambda \left(\omega \eta \rho {∥h_{\mathrm{RD}}∥}^{2}+1-\rho \right)\left(\beta {∥h_{\mathrm{SR}}∥}^{2}+\left(1-\beta \right){∥h_{\mathrm{RD}}∥}^{2}\right)+1}. \end{array}$$

## 3. Performance Analysis

In this work, the wiretap codes for confidential information are assumed the fixed-rate Wyner coded strategy. $R_{s}$ is the confidential information rate and $R_{0}$ is the codeword transmission rate. To protect the information from eavesdropping, the redundancy rate should be positive, denoted by $R_{e}=R_{0}-R_{s}$. On the contrary, when the channel capacity of the untrusted relay *R* is higher than redundancy rate, the information is unsafe, which is the definition of SOP. The SOP can be formulated as follows [53].
(13)$$\begin{array}{l} P_{SOP}^{i}=\mathrm{Pr}\left[I_{R}^{i}>R_{e}\right], \end{array}$$
where $I_{R}^{i}=\frac{1}{2}\log_{2}\left(1+\gamma_{R}^{i}\right)$ and *i* is TAS-RAS scheme, MRT-RAS scheme, TAS-MRC scheme or MRT-MRC scheme. The coefficient 1/2 is due to two phases.

If the channel capacity of *D* is lower than $R_{0}$, *D* is unable to decode the information correctly, and this is the definition of COP. The COP can be formulated as follows [53].
(14)$$\begin{array}{l} P_{COP}^{i}=\mathrm{Pr}\left[I_{D}^{i}<R_{0}\right], \end{array}$$
where $I_{D}^{i}=\frac{1}{2}\log_{2}\left(1+\gamma_{D}^{i}\right)$.

SOP and COP can be used to measure the security and reliability performance, respectively. Furthermore, to comprehensively characterize the security and reliability performance, the EST is proposed in [40]. The EST is defined as the product of the secrecy rate and the probability of a successful transmission, which is given by
(15)$$\begin{array}{l} \zeta^{i}=R_{s}\;\mathrm{Pr}\left[I_{R}^{i}<R_{e},I_{D}^{i}>R_{0}\right]. \end{array}$$

Next, we derive the exactly equations of the SOP, COP and EST of the aforementioned four transmission and reception schemes.

### 3.1. TAS-RAS Scheme

For the TAS-RAS scheme, let $X_{1}={\left|h_{SR}^{*}\right|}^{2}$ and $Y_{1}={\left|h_{RD}^{*}\right|}^{2}$. Substituting Equation (Equation 1) into Equation (Equation 13), the SOP can be calculated as
(16)$$\begin{array}{ll} P_{sop}^{TAS-RAS} & =\mathrm{Pr}\left[I_{R}^{TAS-RAS}>R_{e}\right]\\ & =\mathrm{Pr}\left[\frac{1}{2}\log_{2}\left(1+\frac{\left(1-\rho \right)\beta \lambda {\left|h_{SR}^{*}\right|}^{2}}{\left(1-\rho \right)\left(1-\beta \right)\lambda {\left|h_{RD}^{*}\right|}^{2}+1}\right)>R_{e}\right]\\ & =\int_{0}^{\infty }\int_{X_{1b}}^{\infty }f_{X_{1}}\;\left(x_{1}\right)\;f_{Y_{1}}\;\left(y_{1}\right)dx_{1}dy_{1}, \end{array}$$
where $X_{1b}=\frac{\left(1-\beta \right)t_{1}Y_{1}}{\beta }+\frac{t_{1}}{\left(1-\rho \right)\beta \lambda }$ and $t_{1}=2^{2\left(R_{0}-R_{s}\right)}-1$.

Firstly, the preliminary of the following lemma is present.

**Lemma** **1.**
*The cumulative distribution function (CDF) of $X_{1}$ is given by*
(17)$$\begin{array}{l} F_{X_{1}}\left(x_{1}\right)={\left(1-e^{-\frac{x_{1}}{{\bar{\gamma }}_{SR}}}\right)}^{N_{s}}. \end{array}$$


**Proof.** By taking the derivation operation of $X_{1}$, the probability density function (PDF) of $X_{1}$ is calculated as
(18)$$\begin{array}{l} f_{X_{1}}\left(x_{1}\right)=\frac{N_{s}}{{\bar{\gamma }}_{SR}}e^{-\frac{x_{1}}{{\bar{\gamma }}_{SR}}}{\left(1-e^{-\frac{x_{1}}{{\bar{\gamma }}_{SR}}}\right)}^{N_{s}-1}. \end{array}$$ □

Similarly, the PDF of $Y_{1}$ is calculated as follows.
(19)$$\begin{array}{l} f_{Y_{1}}\left(y_{1}\right)=\frac{N_{d}}{{\bar{\gamma }}_{RD}}e^{-\frac{y_{1}}{{\bar{\gamma }}_{RD}}}{\left(1-e^{-\frac{y_{1}}{{\bar{\gamma }}_{RD}}}\right)}^{N_{d}-1}. \end{array}$$

Substituting Equations (Equation 18) and (Equation 19) into Equation (Equation 16), the SOP can be calculated as
(20)$$\begin{array}{l} P_{sop}^{TAS-RAS}=\frac{N_{s}N_{d}}{{\bar{\gamma }}_{SR}{\bar{\gamma }}_{RD}}\int_{0}^{\infty }\int_{X_{1b}}^{\infty }e^{-\left(\frac{x_{1}}{{\bar{\gamma }}_{SR}}+\frac{y_{1}}{{\bar{\gamma }}_{RD}}\right)}\;{\left(1-e^{-\frac{x_{1}}{{\bar{\gamma }}_{SR}}}\right)}^{N_{s}-1}{\left(1-e^{-\frac{y_{1}}{{\bar{\gamma }}_{RD}}}\right)}^{N_{d}-1}dx_{1}dy_{1}. \end{array}$$

After some manipulations, the SOP expression of the TAS-RAS scheme can be rewritten as
(21)$$\begin{array}{l} P_{sop}^{TAS-RAS}=1-\sum_{n_{s}=0}^{N_{s}}\sum_{n_{d}=0}^{N_{d}-1}\left(\begin{array}{c} \;N_{s}\;\\ \;\;n_{s}\; \end{array}\right)\left(\begin{array}{c} \;N_{d}-1\;\\ \;\;\;\;n_{d}\; \end{array}\right)\frac{{\left(-1\right)}^{n_{s}+n_{d}}N_{d}\;\beta \;{\bar{\gamma }}_{SR}\;e^{-\frac{t_{1}n_{s}}{\left(1-\rho \right)\beta \lambda {\bar{\gamma }}_{SR}}}}{\left(1-\beta \right)t_{1}n_{s}{\bar{\gamma }}_{RD}+\left(n_{d}+1\right)\beta {\bar{\gamma }}_{SR}}. \end{array}$$

From Equation (Equation 21), the SOP of the TAS-RAS scheme is closely related to β and ρ. In particular, the SOP is improved when β increases from 0 to 1. This is because the allocated power for transmitting information increases when gradually increasing β, which in turn leads to the descent of transmit power for the jamming signal. Besides, the security of the system improves when ρ continuously increases. The reason is that *R* extracts less information from the source signal when the portion of harvested energy increases.

Substituting Equation (Equation 3) into Equation (Equation 14), the COP of the TAS-RAS scheme is given by
(22)$$\begin{array}{ll} P_{cop}^{TAS-RAS} & =\mathrm{Pr}\left[I_{D}^{TAS-RAS}<R_{0}\right]\\ & \approx \mathrm{Pr}\left[\frac{\omega \eta \rho \left(1-\rho \right)\beta \lambda {\left|h_{SR}^{*}\right|}^{2}{\left|h_{RD}^{*}\right|}^{2}}{\omega \eta \rho {\left|h_{RD}^{*}\right|}^{2}+1-\rho }<t_{2}\right]\\ & =\int_{0}^{\infty }\int_{0}^{X_{1s}}f_{X_{1}}\;\left(x_{1}\right)\;f_{Y_{1}}\;\left(y_{1}\right)dx_{1}dy_{1}, \end{array}$$
where $X_{1s}=\frac{t_{2}}{\left(1-\rho \right)\beta \lambda }+\frac{t_{2}}{\omega \eta \rho \beta \lambda Y_{1}}$, $t_{2}=2^{2R_{0}}-1$.

The approximation follows the fact that when λ is sufficiently high compared with the transmit power and channel gains, the sum item of the denominator in Equation (Equation 3) can be negligible and the denominator can be approximated as $\lambda \left(\omega \eta \rho {\left|h_{RD}^{*}\right|}^{2}+1-\rho \right)\left(\beta {\left|h_{SR}^{*}\right|}^{2}+\left(1-\beta \right){\left|h_{RD}^{*}\right|}^{2}\right)$ [15,31,54].

Substituting Equations (Equation 18) and (Equation 19) into Equation (Equation 22), the COP of the TAS-RAS scheme can be rewritten as
(23)$$\begin{array}{l} P_{cop}^{TAS-RAS}\;=\;\frac{N_{d}}{{\bar{\gamma }}_{RD}}\sum_{n_{s}=0}^{\;N_{s}\;}\sum_{n_{d}=0}^{\;N_{d}\;-\;1\;}\;\left(\begin{array}{c} \;\;N_{s}\;\;\\ \;\;\;n_{s}\;\; \end{array}\right)\left(\begin{array}{c} \;\;\;N_{d}\;-\;1\;\;\;\\ \;\;\;\;\;n_{d}\;\; \end{array}\right){\left(-1\right)}^{n_{s}+n_{d}}\;e^{-\frac{n_{s}t_{2}}{\left(1-\rho \right)\beta \lambda {\bar{\gamma }}_{SR}}}\;\int_{0}^{\infty }\;\;\;e^{-\left(\frac{n_{s}t_{2}}{\omega \eta \rho \beta \lambda {\bar{\gamma }}_{SR}y_{1}}\;+\;\frac{\left(n_{d}+1\right)y_{1}}{{\bar{\gamma }}_{RD}}\right)}\;\;dy_{1}. \end{array}$$

Applying Equation (3.324.1) in [55], the COP can be given by
(24)$$\begin{array}{l} P_{cop}^{TAS-RAS}\;=\;\;\;\sum_{n_{s}=0}^{N_{s}}\sum_{n_{d}=0}^{N_{d}-1}\;\;\left(\begin{array}{c} \;\;\;\;N_{s}\;\;\\ \;\;\;\;n_{s}\;\;\; \end{array}\right)\;\;\left(\begin{array}{c} \;\;\;\;N_{d}\;-\;1\;\;\;\;\\ \;\;\;\;\;\;\;n_{d}\;\;\;\; \end{array}\right){\left(-1\right)}^{n_{s}+n_{d}}e^{-\frac{n_{s}\;t_{2}}{\left(1-\rho \right)\beta \lambda {\bar{\gamma }}_{SR}}}\sqrt{\frac{4n_{s}t_{2}N_{d}^{2}}{\omega \eta \rho \beta \lambda {\bar{\gamma }}_{SR}{\bar{\gamma }}_{RD}\left(n_{d}+1\right)}}K_{1}\left(\sqrt{\frac{4n_{s}t_{2}\left(n_{d}+1\right)}{\omega \eta \rho \beta \lambda {\bar{\gamma }}_{SR}{\bar{\gamma }}_{RD}}}\right), \end{array}$$
where $K_{1}\left(•\right)$ is the second kind of Modified Bessel function.

Equation (Equation 24) shows the COP decreases with continuously increasing β. The probability of decoding the confidential information improves when the allocated power of *S* increases.

Based on the expressions of SOP and COP, it can be noted that increasing λ improves the reliability of the system. Nevertheless, it also leads to increasing the risk of information leakage. Therefore, the EST is used for comprehensively measuring the reliability and security in untrusted energy harvesting networks. Based on Equation (Equation 15), the EST is given by
(25)$$\begin{array}{ll} \zeta^{TAS-RAS} & =\frac{R_{s}}{2}\mathrm{Pr}\left[I_{R}^{TAS-RAS}<R_{e},I_{D}^{TAS-RAS}>R_{0}\right]\\ & \approx \frac{R_{s}}{2}\mathrm{Pr}\left[\frac{\left(1-\rho \right)\beta \lambda {\left|h_{SR}^{*}\right|}^{2}}{\left(1-\rho \right)\left(1-\beta \right)\lambda {\left|h_{RD}^{*}\right|}^{2}+1}<t_{1},\right.\left.\frac{\omega \eta \rho \left(1-\rho \right)\beta \lambda {\left|h_{SR}^{*}\right|}^{2}{\left|h_{RD}^{*}\right|}^{2}}{\omega \eta \rho {\left|h_{RD}^{*}\right|}^{2}+1-\rho }>t_{2}\right]\\ & =\frac{R_{s}}{2}\mathrm{Pr}\left[X_{1}<X_{1b},X_{1}>X_{1s}\right]. \end{array}$$

Note that $X_{1b}$ is larger than $X_{1s}$ only when $Y_{1}>u$, where $u=\left(b+\sqrt{b^{2}+4ac}\right)/2a$, $a=\omega \eta \rho \lambda \left(1-\rho \right)\left(1-\beta \right)t_{1}$, $b=\omega \eta \rho (t_{2}-t_{1})$, $c=\left(1-\rho \right)t_{2}$.

Hence, $\zeta^{TAS-RAS}$ can be calculated as
(26)$$\begin{array}{ll} \zeta^{TAS-RAS} & =\frac{R_{s}}{2}\int_{u}^{\infty }\int_{X_{1s}}^{X_{1b}}f_{X_{1}}\;\left(x_{1}\right)\;f_{Y_{1}}\;\left(y_{1}\right)dx_{1}dy_{1}\\ & =\frac{R_{s}N_{d}}{2{\bar{\gamma }}_{RD}}\left\{\int_{u}^{\infty }\;e^{-\frac{y_{1}}{{\bar{\gamma }}_{RD}}}\;{\left(1-e^{-\frac{y_{1}}{{\bar{\gamma }}_{RD}}}\right)}^{N_{d}-1}\left({\left(1\;-\;e^{\;-\;\frac{X_{1b}}{{\bar{\gamma }}_{SR}}}\right)}^{N_{s}}\;-\;{\left(1-e^{\;-\;\frac{X_{1s}}{{\bar{\gamma }}_{SR}}}\right)}^{N_{s}}\right)dy_{1}\right\}. \end{array}$$

Applying Equations (1.111) and (3.381.4) in [55], the EST expression is given by
(27)$$\begin{array}{l} \zeta^{TAS-RAS}=\frac{R_{s}N_{d}}{2}\sum_{n_{s}=0}^{N_{s}}\sum_{n_{d}=0}^{N_{d}-1}\left(\begin{array}{c} N_{s}\\ \;n_{s} \end{array}\right)\left(\begin{array}{c} N_{d}-1\\ \;\;\;n_{d} \end{array}\right){\left(-1\right)}^{n_{s}+n_{d}}\\ \times \left\{\frac{\beta {\bar{\gamma }}_{SR}e^{-u_{1}}}{\left(1-\beta \right)t_{1}n_{s}{\bar{\gamma }}_{RD}+\left(n_{d}+1\right)\beta {\bar{\gamma }}_{SR}}-\sum_{m=0}^{\infty }\frac{{\left(-t_{2}n_{s}\left(n_{d}+1\right)\right)}^{m}e^{-\;\frac{t_{2}n_{s}}{\left(1-\rho \right)\beta \lambda {\bar{\gamma }}_{SR}}}\;\Gamma \left(1-m,\frac{(n_{d}+1)u}{{\bar{\gamma }}_{RD}}\right)}{\left(n_{d}+1\right)m!\;{\left(\omega \eta \rho \beta \lambda {\bar{\gamma }}_{SR}{\bar{\gamma }}_{RD}\right)}^{m}}\right\}, \end{array}$$
where $u_{1}=\left[\left(1-\beta \right)\left(1-\rho \right)\lambda t_{1}n_{s}u{\bar{\gamma }}_{RD}+\beta \left(1-\rho \right)\left(n_{d}+1\right)\lambda u{\bar{\gamma }}_{SR}+t_{1}n_{s}{\bar{\gamma }}_{RD}\right]/\left[\left(1-\rho \right)\lambda \beta {\bar{\gamma }}_{SR}{\bar{\gamma }}_{RD}\right]$.

Based on Equations (Equation 21), (Equation 24) and (Equation 27), there are optimal values of β and ρ that maximize the EST, as shown in Figures 4 and 7, respectively.

When λ goes to infinity, the asymptotical performance of the EST is considered. According to Equation (Equation 25) and after some manipulations, the asymptotic approximation analysis of the EST in the TAS-RAS scheme can be derived as
(28)$$\begin{array}{ll} \lim_{\lambda \to \infty }\zeta^{TAS-RAS} & =\lim_{\lambda \to \infty }\frac{R_{s}}{2}\mathrm{Pr}\left[X_{1}<\frac{\left(1-\beta \right)t_{1}y}{\beta }\right]\\ & =\frac{R_{s}N_{d}}{2}\sum_{n_{s}=0}^{N_{s}}\sum_{n_{d}=0}^{N_{d}-1}\left(\begin{array}{c} N_{s}\\ \;n_{s} \end{array}\right)\left(\begin{array}{c} N_{d}-1\\ \;\;\;n_{d} \end{array}\right)\frac{{\left(-1\right)}^{n_{s}+n_{d}}\beta {\bar{\gamma }}_{SR}}{\left(1-\beta \right)t_{1}n_{s}{\bar{\gamma }}_{RD}+\left(n_{d}+1\right)\beta {\bar{\gamma }}_{SR}}. \end{array}$$

Equation (Equation 28) shows that the EST of the TAS-RAS scheme is independent of the transmit SNR at high transmit SNR. Hence, the EST should be a constant when λ→∞.

### 3.2. MRT-RAS Scheme

In the MRT-RAS scheme, substituting Equation (Equation 4) into Equation (Equation 13), the SOP can be rewritten as
(29)$$\begin{array}{ll} P_{sop}^{MRT-RAS} & =\mathrm{Pr}\left[I_{R}^{MRT-RAS}>R_{e}\right]\\ & =\mathrm{Pr}\left[\frac{1}{2}\log_{2}\left(1+\frac{\left(1-\rho \right)\beta \lambda {∥h_{\mathrm{SR}}∥}^{2}}{\left(1-\rho \right)\left(1-\beta \right)\lambda {\left|h_{RD}^{*}\right|}^{2}+1}\right)>R_{e}\right]\\ & =\int_{0}^{\infty }\int_{X_{1b}}^{\infty }f_{X_{2}}\;\left(x_{2}\right)\;f_{Y_{1}}\;\left(y_{1}\right)dx_{2}dy_{1}, \end{array}$$
where $X_{2}={∥h_{\mathrm{SR}}∥}^{2}$, and the PDF of $X_{2}$ can be written as
(30)$$\begin{array}{l} f_{X_{2}}\left(x_{2}\right)=\frac{x_{2}^{N_{s}-1}\;e^{-\frac{x_{2}}{{\bar{\gamma }}_{SR}}}}{\Gamma \left(N_{s}\right){\bar{\gamma }}_{SR}^{N_{s}}}. \end{array}$$

Submitting Equations (Equation 19) and (Equation 30) into Equation (Equation 29), the SOP can be expressed as
(31)$$\begin{array}{l} P_{sop}^{MRT-RAS}=\frac{N_{d}}{\Gamma \left(N_{s}\right){\bar{\gamma }}_{SR}^{N_{s}}{\bar{\gamma }}_{RD}}\int_{0}^{\infty }\int_{X_{1b}}^{\infty }x_{2}^{N_{s}-1}e^{-\left(\frac{x_{2}}{{\bar{\gamma }}_{SR}}+\frac{y_{1}}{{\bar{\gamma }}_{RD}}\right)}{\left(1-e^{-\frac{y_{1}}{{\bar{\gamma }}_{RD}}}\right)}^{N_{d}-1}dx_{2}dy_{1}. \end{array}$$

Hence, the SOP of the MRT-RAS scheme can be calculated as
(32)$$\begin{array}{l} P_{sop}^{MRT-RAS}=\sum_{n_{s}=0}^{N_{s}-1}\sum_{n_{d}=0}^{N_{d}-1}\sum_{k=0}^{n_{s}}\left(\begin{array}{c} \;\;N_{d}-1\;\;\\ \;\;\;\;\;n_{d}\;\; \end{array}\right)\left(\begin{array}{c} \;\;n_{s}\\ \;\;\;\;\;\;k\;\; \end{array}\right)\frac{{\left(-1\right)}^{n_{d}}N_{d}\;k!\;t_{1}^{n_{s}}{\left(\left(1-\beta \right){\bar{\gamma }}_{RD}\right)}^{k}e^{-\frac{t_{1}}{\left(1-\rho \right)\beta \lambda {\bar{\gamma }}_{SR}}}}{n_{s}!{\left(\beta {\bar{\gamma }}_{SR}\right)}^{n_{s}-k-1}{\left((1-\rho )\lambda \right)}^{n_{s}-k}u_{2}^{k+1}}. \end{array}$$
where $u_{2}=\left(1-\beta \right)t_{1}{\bar{\gamma }}_{RD}+\left(n_{d}+1\right)\beta {\bar{\gamma }}_{SR}$. The SOP of the MRT-RAS scheme is associated with parameters β and ρ. Moreover, the SOP increases when β increases, and it decreases as ρ increases.

Plugging Equation (Equation 6) into Equation (Equation 14), the COP of the MRT-RAS scheme is given by
(33)$$\begin{array}{ll} P_{cop}^{MRT-RAS} & =\mathrm{Pr}\left[I_{D}^{MRT-RAS}<R_{0}\right]\\ & \approx \mathrm{Pr}\left[\frac{\omega \eta \rho \left(1-\rho \right)\beta \lambda {∥h_{\mathrm{SR}}∥}^{2}{\left|h_{RD}^{*}\right|}^{2}}{\omega \eta \rho {\left|h_{RD}^{*}\right|}^{2}+1-\rho }<t_{2}\right]\\ & =\int_{0}^{\infty }\int_{0}^{X_{1s}}f_{X_{2}}\;\left(x_{2}\right)\;f_{Y_{1}}\;\left(y_{1}\right)dx_{2}dy_{1}. \end{array}$$

Substituting Equations (Equation 19) and (Equation 30) into Equation (Equation 33), the COP can be written as
(34)$$\begin{array}{l} P_{cop}^{MRT-RAS}=1\;\;-\;\;N_{d}\sum_{n_{s}=0}^{N_{s}-1}\sum_{n_{d}=0}^{N_{d}-1}\sum_{m=0}^{n_{s}}\left(\begin{array}{c} \;\;N_{d}\;-\;1\;\;\\ \;\;\;\;\;n_{d}\;\; \end{array}\right)\left(\begin{array}{c} \;\;n_{s}\;\;\\ \;\;m\;\; \end{array}\right)\frac{{\left(-1\right)}^{n_{d}}t_{2}^{n_{s}}e^{-\frac{t_{2}}{\left(1-\rho \right)\beta \lambda {\bar{\gamma }}_{SR}}}\int_{0}^{\infty }y_{1}^{m}e^{-\left(\frac{t_{2}}{\omega \eta \rho \beta \lambda {\bar{\gamma }}_{SR}y_{1}}\;+\;\frac{\left(n_{d}+1\right)y_{1}}{{\bar{\gamma }}_{RD}}\right)\;}}{{\bar{\gamma }}_{RD}\;n_{s}!\;{\left(1-\rho \right)}^{n_{s}-m}{\left(\beta \lambda {\bar{\gamma }}_{SR}\right)}^{n_{s}}{\left(\omega \eta \rho \right)}^{m}}. \end{array}$$

Utilizing Equation (3.471.9) in [55], the expression of COP is given by
(35)$$\begin{array}{l} P_{cop}^{MRT-RAS}=1\;-\;\frac{2N_{d}\sum_{n_{s}=0}^{N_{s}-1}\sum_{n_{d}=0}^{N_{d}-1}\sum_{m=0}^{n_{s}}\left(\begin{array}{c} \;\;N_{d}-1\;\;\\ \;\;\;\;\;n_{d}\;\; \end{array}\right)\left(\begin{array}{c} \;\;n_{s}\;\;\\ \;\;m\;\; \end{array}\right){\left(-1\right)}^{n_{d}}t_{2}^{\frac{2n_{s}+1-m}{2}}e^{-\frac{t_{2}}{\left(1-\rho \right)\beta \lambda {\bar{\gamma }}_{SR}}}K_{1-m}\left(\sqrt{\frac{4t_{2}\left(n_{d}+1\right)}{\omega \eta \rho \beta \lambda {\bar{\gamma }}_{SR}{\bar{\gamma }}_{RD}}}\right)}{n_{s}!{\left(1-\rho \right)}^{n_{s}-m}{\left(\beta \lambda {\bar{\gamma }}_{SR}\right)}^{\frac{2n_{s}+1-m}{2}}{\left(\omega \eta \rho {\bar{\gamma }}_{RD}\right)}^{\frac{m+1}{2}}{\left(n_{d}+1\right)}^{\frac{1-m}{2}}}. \end{array}$$

Equation (Equation 35) shows that the COP of the MRT-RAS scheme decreases with the increase of β. According to Equation (Equation 15), the closed-form EST expression is calculated as follows.
(36)$$\begin{array}{ll} \zeta^{MRT-RAS} & =\frac{R_{s}}{2}\mathrm{Pr}\left[I_{R}^{MRT-RAS}<R_{e},I_{D}^{MRT-RAS}>R_{0}\right]\\ & \approx \frac{R_{s}}{2}\mathrm{Pr}\left[\frac{\left(1-\rho \right)\beta \lambda {∥h_{\mathrm{SR}}∥}^{2}}{\left(1-\rho \right)\left(1-\beta \right)\lambda {\left|h_{RD}^{*}\right|}^{2}+1}<t_{1},\right.\left.\frac{\omega \eta \rho \left(1-\rho \right)\beta \lambda {∥h_{\mathrm{SR}}∥}^{2}{\left|h_{RD}^{*}\right|}^{2}}{\omega \eta \rho {∥h_{\mathrm{RD}}∥}^{2}+1-\rho }>t_{2}\right]\\ & =\frac{R_{s}}{2}\int_{u}^{\infty }\int_{X_{1s}}^{X_{1b}}f_{X_{2}}\;\left(x_{2}\right)\;f_{Y_{1}}\;\left(y_{1}\right)dx_{2}dy_{1}. \end{array}$$

Substituting Equations (Equation 19) and (Equation 30) into Equation (Equation 36), the EST can be rewritten as
(37)$$\begin{array}{ll} \zeta^{MRT-RAS} & =\frac{R_{s}N_{d}}{2{\bar{\gamma }}_{RD}}\left\{\int_{u}^{\infty }\sum_{n_{s}=0}^{N_{S}-1}\frac{1}{n_{s}!}{\left(\frac{X_{1s}}{{\bar{\gamma }}_{SR}}\right)}^{n_{s}}e^{-\left(\frac{X_{1s}}{{\bar{\gamma }}_{SR}}+\;\frac{y_{1}}{{\bar{\gamma }}_{RD}}\right)}{\left(1-e^{-\frac{y_{1}}{{\bar{\gamma }}_{RD}}}\right)}^{N_{d}-1}dy_{1}\right.\\ & -\left.\int_{u}^{\infty }\sum_{n_{s}=0}^{N_{S}-1}\frac{1}{n_{s}!}{\left(\frac{X_{1b}}{{\bar{\gamma }}_{SR}}\right)}^{n_{s}}e^{-\;\left(\frac{X_{1b}}{{\bar{\gamma }}_{SR}}+\;\frac{y_{1}}{{\bar{\gamma }}_{RD}}\right)}{\left(1-e^{-\frac{y_{1}}{{\bar{\gamma }}_{RD}}}\right)}^{N_{d}-1}dy_{1}\right\}\\ & =\frac{R_{s}N_{d}}{2}\left\{\Xi_{1}-\Xi_{2}\right\}. \end{array}$$

Applying Equation (3.381.3.8) in [55], $\Xi_{1}$ and $\Xi_{2}$, respectively, are as follows.
(38)$$\begin{array}{l} \Xi_{1}=\sum_{n_{s}=0}^{N_{s}-1}\sum_{n_{d}=0}^{N_{d}-1}\sum_{m=0}^{n_{s}}\sum_{k=0}^{\infty }\left(\begin{array}{c} \;\;\;N_{d}-1\;\;\;\\ \;\;\;\;\;\;n_{d}\;\;\; \end{array}\right)\left(\begin{array}{c} \;\;\;n_{s}\;\;\;\\ \;\;\;m\;\;\; \end{array}\right)\frac{{\left(-1\right)}^{n_{d}+k}t_{2}^{n_{s}+k}{\left(n_{d}+1\right)}^{m+k-1}e^{-\frac{t_{2}}{\left(1-\rho \right)\beta \lambda {\bar{\gamma }}_{SR}}}\Gamma \left(1-m-k,\frac{(n_{d}+1)u}{{\bar{\gamma }}_{RD}}\right)}{n_{s}!\;k!\;{\left(1-\rho \right)}^{n_{s}-m}{\left(\beta \lambda {\bar{\gamma }}_{SR}\right)}^{n_{s}+k}{\left(\omega \eta \rho {\bar{\gamma }}_{RD}\right)}^{m+k}}, \end{array}$$
(39)$$\begin{array}{l} \Xi_{2}=\sum_{n_{s}=0}^{N_{s}-1}\sum_{n_{d}=0}^{N_{d}-1}\sum_{m=0}^{n_{s}}\left(\begin{array}{c} \;\;\;N_{d}-1\;\;\;\\ \;\;\;\;\;\;n_{d}\;\;\; \end{array}\right)\left(\begin{array}{c} \;\;\;n_{s}\;\;\;\\ \;\;\;m\;\;\; \end{array}\right)\frac{{\left(-1\right)}^{n_{d}}t_{1}^{n_{s}}{\left(\left(1-\beta \right){\bar{\gamma }}_{RD}\right)}^{m}e^{-\;\frac{t_{1}}{\left(-\rho \right)\beta \lambda {\bar{\gamma }}_{SR}}}\Gamma \left(1+m,\frac{u_{2}u}{\beta \;\bar{\gamma }{}_{SR}{\bar{\gamma }}_{RD}}\right)}{n_{s}!\;u_{2}^{m+1}{\left((1-\rho )\lambda \right)}^{n_{s}-m}{\left(\beta \;{\bar{\gamma }}_{SR}\right)}^{n_{s}-m-1}}. \end{array}$$

The maximal EST with the optimal β and ρ is illustrated in Figures 4 and 7, respectively.

Based on Equations (Equation 37), (Equation 38) and (Equation 39), when λ goes to infinity, the asymptotical performance of the EST in the MRT-RAS scheme can be calculated as
(40)$$\begin{array}{l} \lim_{\lambda \to \infty }\zeta^{MRT-RAS}=\frac{R_{s}}{2}\left(1-\sum_{n_{s}=0}^{N_{s}-1}\sum_{n_{d}=0}^{N_{d}-1}\left(\begin{array}{c} \;\;N_{d}-1\;\;\\ \;\;\;\;\;n_{d}\;\; \end{array}\right)\;\right.\left.\frac{{\left(-1\right)}^{n_{d}}N_{d}\;\beta {\bar{\gamma }}_{SR}{\left(\left(1-\beta \right)t_{1}{\bar{\gamma }}_{RD}\right)}^{n_{s}}}{{\left(\left(1-\beta \right)t_{1}{\bar{\gamma }}_{RD}+\left(n_{d}+1\right)\beta {\bar{\gamma }}_{SR}\right)}^{n_{s}+1}}\right). \end{array}$$

Equation (Equation 40) shows that the EST of the MRT-RAS scheme is irrelevant with the transmit SNR at high transmit SNR.

### 3.3. TAS-MRC Scheme

In TAS-MRC scheme, plugging Equation (Equation 7) into Equation (Equation 13), the SOP can be rewritten as
(41)$$\begin{array}{ll} P_{sop}^{TAS-MRC} & =\mathrm{Pr}\left[I_{R}^{TAS-MRC}>R_{e}\right]\\ & =\mathrm{Pr}\left[\;\frac{1}{2}\log_{2}\left(1\;+\;\frac{\left(1-\rho \right)\beta \lambda {\left|h_{SR}^{*}\right|}^{2}}{\left(1-\rho \right)\left(1-\beta \right)\lambda {∥h_{\mathrm{RD}}∥}^{2}+1}\right)\;>\;R_{e}\;\right]\\ & =\int_{0}^{\infty }\int_{X_{2b}}^{\infty }f_{X_{1}}\;\left(x_{1}\right)\;f_{Y_{2}}\;\left(y_{2}\right)dx_{1}dy_{2}, \end{array}$$
where $X_{2b}=\frac{\left(1-\beta \right)t_{1}Y_{2}}{\beta }+\frac{t_{1}}{\left(1-\rho \right)\beta \lambda }$.

Let $Y_{2}={∥h_{\mathrm{RD}}∥}^{2}$, then the PDF of $Y_{2}$ is given by
(42)$$\begin{array}{l} f_{Y_{2}}\left(y_{2}\right)=\frac{y_{2}^{N_{d}-1}\;e^{-\frac{y_{2}}{{\bar{\gamma }}_{RD}}}}{{\bar{\gamma }}_{RD}^{N_{d}}\;\;\Gamma \left(N_{d}\right)}. \end{array}$$

Substituting Equations (Equation 18) and (Equation 42) into Equation (Equation 41), the SOP can be calculated as
(43)$$\begin{array}{l} P_{sop}^{TAS-MRC}=1-\sum_{n_{s}=0}^{N_{s}}\left(\begin{array}{c} N_{s}\\ \;n_{s} \end{array}\right)\frac{{\left(-1\right)}^{n_{s}}e^{-\frac{t_{1}n_{s}}{\left(1-\rho \right)\beta \lambda {\bar{\gamma }}_{SR}}}\int_{0}^{\infty }y_{2}^{N_{d}-1}e^{-\frac{\left(1-\beta \right)t_{1}\;n_{s}{\bar{\gamma }}_{RD}+\beta {\bar{\gamma }}_{SR}}{\beta \;{\bar{\gamma }}_{SR}{\bar{\gamma }}_{RD}}y_{2}}dy_{2}}{{\bar{\gamma }}_{RD}^{N_{d}}\Gamma \left(N_{d}\right){\left(\left(1-\beta \right)t_{1}n_{s}{\bar{\gamma }}_{RD}+\beta {\bar{\gamma }}_{SR}\right)}^{N_{d}}}. \end{array}$$

Using Equation (3.35.3) in [55], the expression of SOP can be calculated as
(44)$$\begin{array}{l} P_{sop}^{TAS-MRC}=1-\sum_{n_{s}=0}^{N_{s}}\left(\begin{array}{c} N_{s}\\ \;n_{s} \end{array}\right)\frac{{\left(-1\right)}^{n_{s}}{\left(\beta {\bar{\gamma }}_{SR}\right)}^{N_{d}}e^{-\frac{t_{1}n_{s}}{\left(1-\rho \right)\beta \lambda {\bar{\gamma }}_{SR}}}}{{\left(\left(1-\beta \right)t_{1}n_{s}{\bar{\gamma }}_{RD}+\beta {\bar{\gamma }}_{SR}\right)}^{N_{d}}}. \end{array}$$

Combining Equations (Equation 9) and (Equation 14), the COP of the TAS-MRC scheme is written as
(45)$$\begin{array}{ll} P_{cop}^{TAS-MRC} & =\mathrm{Pr}\left[I_{D}^{TAS-MRC}<R_{0}\right]\\ & \approx \mathrm{Pr}\left[\frac{\omega \eta \rho \left(1-\rho \right)\beta \lambda {\left|h_{SR}^{*}\right|}^{2}{∥h_{\mathrm{RD}}∥}^{2}}{\omega \eta \rho {∥h_{\mathrm{RD}}∥}^{2}+1-\rho }<t_{2}\right]\\ & =\int_{0}^{\infty }\int_{0}^{X_{2s}}f_{X_{1}}\;\left(x_{1}\right)\;f_{Y_{2}}\;\left(y_{2}\right)dx_{1}dy_{2}, \end{array}$$
where $X_{2s}=\frac{t_{2}}{\left(1-\rho \right)\beta \lambda }+\frac{t_{2}}{\omega \eta \rho \beta \lambda Y_{2}}$.

Substituting Equations (Equation 18) and (Equation 42) into Equation (Equation 45) and utilizing Equation (3.471.9) in [55], the COP can be calculated as
(46)$$\begin{array}{l} P_{cop}^{TAS-MRC}=1+2\sum_{n_{s}=1}^{N_{s}}\left(\begin{array}{c} N_{s}\\ \;n_{s} \end{array}\right)\frac{{\left(-1\right)}^{n_{s}}{\left(t_{2}n_{s}\right)}^{\frac{N_{d}}{2}}e^{-\frac{t_{2}\;n_{s}}{\left(1-\rho \right)\beta \lambda {\bar{\gamma }}_{SR}}}K_{N_{d}}\left(\sqrt{\frac{4t_{2}\;n_{s}}{\omega \eta \rho \beta \lambda {\bar{\gamma }}_{SR}{\bar{\gamma }}_{RD}}}\right)}{\Gamma \left(N_{d}\right){\left(\omega \eta \rho \beta \lambda {\bar{\gamma }}_{SR}{\bar{\gamma }}_{RD}\right)}^{\frac{N_{d}}{2}}}. \end{array}$$

To measure the trade-off between the SOP and COP, the EST can be written as
(47)$$\begin{array}{ll} \zeta^{TAS-MRC} & =\frac{R_{s}}{2}\mathrm{Pr}\left[I_{R}^{TAS-MRC}\;<\;R_{e},I_{D}^{TAS-MRC}\;>\;R_{0}\right]\\ & \approx \frac{R_{s}}{2}\mathrm{Pr}\left[\frac{\left(1-\rho \right)\beta \lambda {\left|h_{SR}^{*}\right|}^{2}}{\left(1-\rho \right)\left(1-\beta \right)\lambda {∥h_{\mathrm{RD}}∥}^{2}+1}<t_{1}\right.\left.\frac{\omega \eta \rho \left(1-\rho \right)\beta \lambda {\left|h_{SR}^{*}\right|}^{2}{∥h_{\mathrm{RD}}∥}^{2}}{\omega \eta \rho {∥h_{\mathrm{RD}}∥}^{2}+1-\rho }>t_{2}\right]\\ & =\frac{R_{s}}{2}\int_{u}^{\infty }\int_{X_{2s}}^{X_{2b}}f_{X_{1}}\;\left(x_{1}\right)\;f_{Y_{2}}\;\left(y_{2}\right)dx_{1}dy_{2}. \end{array}$$

Substituting Equations (Equation 18) and (Equation 42) into Equation (Equation 47), using Equation (3.381.3.8) in [55], and after some manipulations, the EST can be calculated as
(48)$$\begin{array}{ll} \zeta^{TAS-MRC} & =\frac{R_{s}}{2{\bar{\gamma }}_{RD}^{N_{d}}\;\Gamma \left(N_{d}\right)}\int_{u}^{\infty }y_{2}^{N_{d}-1}e^{-\frac{y_{2}}{{\bar{\gamma }}_{RD}}}\times \left({\left(1-e^{-\frac{X_{2b}}{{\bar{\gamma }}_{SR}}}\right)}^{N_{s}}-{\left(1-e^{-\frac{X_{2s}}{{\bar{\gamma }}_{SR}}}\right)}^{N_{s}}\right)dy_{2}\\ & =\frac{R_{s}}{2\Gamma \left(N_{d}\right)}\left\{\Xi_{3}-\Xi_{4}\right\}, \end{array}$$
where $\Xi_{3}$ and $\Xi_{4}$ are as follows.
(49)$$\begin{array}{l} \Xi_{3}=\sum_{n_{s}=0}^{N_{s}}\left(\begin{array}{c} N_{s}\\ \;n_{s} \end{array}\right)\frac{{\left(-1\right)}^{n_{s}}{\left(\beta {\bar{\gamma }}_{SR}\right)}^{N_{d}}e^{-\;\;\frac{t_{1}n_{s}}{\left(1-\rho \right)\beta \lambda {\bar{\gamma }}_{SR}}}\Gamma \left(N_{d},\frac{\left(1-\beta \right)t_{1}n_{s}u{\bar{\gamma }}_{RD}+\beta u{\bar{\gamma }}_{SR}}{\beta {\bar{\gamma }}_{SR}{\bar{\gamma }}_{RD}}\right)}{{\left(\left(1-\beta \right)t_{1}n_{s}{\bar{\gamma }}_{RD}+\beta {\bar{\gamma }}_{SR}\right)}^{N_{d}}}, \end{array}$$
(50)$$\begin{array}{l} \Xi_{4}=\sum_{n_{s}=0}^{N_{s}}\sum_{k=0}^{\infty }\left(\begin{array}{c} N_{s}\\ \;n_{s} \end{array}\right)\frac{{\left(-1\right)}^{n_{s}+k}e^{-\;\;\frac{t_{2}n_{s}}{\left(1-\rho \right)\beta \lambda {\bar{\gamma }}_{SR}}}{\left(t_{2}n_{s}\right)}^{k}\Gamma \left(N_{d}-k,\frac{u}{{\bar{\gamma }}_{RD}}\right)}{k!{\left(\omega \eta \rho \beta \lambda {\bar{\gamma }}_{SR}{\bar{\gamma }}_{RD}\right)}^{k}}. \end{array}$$

Therefore, when λ→∞, the asymptotical performance of the EST in the TAS-MRC scheme can be calculated as
(51)$$\begin{array}{l} \lim_{\lambda \to \infty }\zeta^{TAS-MRC}=\frac{R_{s}}{2}\sum_{n_{s}=0}^{N_{s}}\left(\begin{array}{c} N_{s}\\ n_{s} \end{array}\right)\frac{{\left(-1\right)}^{n_{s}}{\left(\beta {\bar{\gamma }}_{SR}\right)}^{N_{d}}}{{\left(\left(1-\beta \right)t_{1}n_{s}{\bar{\gamma }}_{RD}\;+\;\beta {\bar{\gamma }}_{SR}\right)}^{n_{d}}}. \end{array}$$

Equation (Equation 51) indicates that the EST of the TAS-MRC scheme is a constant at high transmit SNR.

### 3.4. MRT-MRC Scheme

In the MRT-MRC scheme, combining Equations (Equation 10), (Equation 12), (Equation 13) and (Equation 14), the exact expressions of the SOP and COP can be obtained as Equations (Equation 52) and (Equation 53).
(52)$$\begin{array}{ll} P_{sop}^{MRT-MRC} & =\mathrm{Pr}\left[I_{R}^{MRT-MRC}>R_{e}\right]\\ & =\mathrm{Pr}\left[\frac{1}{2}\log_{2}\left(1+\frac{\left(1-\rho \right)\beta \lambda {∥h_{\mathrm{SR}}∥}^{2}}{\left(1-\rho \right)\left(1-\beta \right)\lambda {∥h_{\mathrm{RD}}∥}^{2}+1}\right)>R_{e}\right]\\ & =\sum_{n_{s}=0}^{N_{s}-1}\sum_{m=0}^{n_{1}}\left(\begin{array}{c} n_{s}\\ m \end{array}\right)\frac{\left(N_{d}+m-1\right)!\;\;t_{1}^{n_{s}}{\left(\beta {\bar{\gamma }}_{SR}\right)}^{N_{d}}{\left(\left(1-\beta \right){\bar{\gamma }}_{RD}\right)}^{m}e^{-\frac{t_{1}}{\left(1-\rho \right)\beta \lambda {\bar{\gamma }}_{SR}}}}{n_{s}!\Gamma \left(N_{d}\right){\left(\left(1-\rho \right)\lambda \beta {\bar{\gamma }}_{SR}\right)}^{n_{s}-m}u_{3}^{N_{d}+m}}, \end{array}$$
where $u_{3}=\left(1-\beta \right)t_{1}{\bar{\gamma }}_{RD}+\beta {\bar{\gamma }}_{SR}$.

(53)$$\begin{array}{ll} P_{cop}^{MRT-MRC} & =\mathrm{Pr}\left[I_{D}^{MRT-MRC}<R_{0}\right]\\ & \approx \mathrm{Pr}\left[\frac{\omega \eta \rho \left(1-\rho \right)\beta \lambda {∥h_{\mathrm{SR}}∥}^{2}{∥h_{\mathrm{RD}}∥}^{2}}{\omega \eta \rho {∥h_{\mathrm{RD}}∥}^{2}+1-\rho }<t_{2}\right]\\ & =1-\frac{2}{\Gamma \left(N_{d}\right)}\sum_{n_{s}=1}^{N_{s}-1}\sum_{m=0}^{n_{s}}\left(\begin{array}{c} n_{s}\\ m \end{array}\right)\frac{{\left(t_{2}\right)}^{\frac{2n_{s}+N_{d}-m}{2}}e^{-\frac{t_{2}}{\left(1-\rho \right)\beta \lambda {\bar{\gamma }}_{SR}}}K_{N_{d}-m}\left(\sqrt{\frac{4t_{2}}{\omega \eta \rho \beta \lambda {\bar{\gamma }}_{SR}{\bar{\gamma }}_{RD}}}\right)}{n_{s}!{\left(\left(1-\rho \right)\beta \lambda {\bar{\gamma }}_{SR}\right)}^{n_{s}-m}{\left(\omega \eta \rho \beta \lambda {\bar{\gamma }}_{SR}{\bar{\gamma }}_{RD}\right)}^{\frac{N_{d}+m}{2}}}. \end{array}$$

Equation (Equation 53) shows that the COP decreases when β increases. When much power is allocated for the source signal, the probability of successfully decoding the source information increases.

The EST of the MRT-MRC scheme is calculated as
(54)$$\begin{array}{ll} \zeta^{MRT-MRC} & =\frac{R_{s}}{2}\mathrm{Pr}\left[I_{R}^{MRT-MRC}<R_{e},I_{D}^{MRT-MRC}>R_{0}\right]\\ & \approx \frac{R_{s}}{2}\mathrm{Pr}\left[\frac{\left(1-\rho \right)\beta \lambda {∥h_{\mathrm{SR}}∥}^{2}}{\left(1-\rho \right)\left(1-\beta \right)\lambda {∥h_{\mathrm{RD}}∥}^{2}+1}<t_{1}\right.,\left.\frac{\omega \eta \rho \left(1-\rho \right)\beta \lambda {∥h_{\mathrm{SR}}∥}^{2}{∥h_{\mathrm{RD}}∥}^{2}}{\omega \eta \rho {∥h_{\mathrm{RD}}∥}^{2}+1-\rho }>t_{2}\right]\\ & =\frac{R_{s}}{2\Gamma \left(N_{d}\right)}\left\{\Xi_{5}-\Xi_{6}\right\}, \end{array}$$
where $\Xi_{5}$ and $\Xi_{6}$ can be calculated as Equations (Equation 55) and (Equation 56), respectively.

(55)$$\begin{array}{l} \Xi_{5}=\sum_{n_{s}=0}^{N_{s}-1}\sum_{m=0}^{n_{s}}\sum_{k=0}^{\infty }\left(\begin{array}{c} n_{s}\\ m \end{array}\right)\frac{{\left(-1\right)}^{k}t_{2}^{n_{s}+k}e^{-\;\;\frac{t_{2}}{\left(1-\rho \right)\beta \lambda {\bar{\gamma }}_{SR}}}\Gamma \left(N_{d}-m-k,\frac{u}{{\bar{\gamma }}_{RD}}\right)}{n_{s}!\;k!{\left(\left(1-\rho \right)\beta \lambda {\bar{\gamma }}_{SR}\right)}^{n_{s}-m}{\left(\omega \eta \rho \beta \lambda {\bar{\gamma }}_{SR}{\bar{\gamma }}_{RD}\right)}^{m+k}}, \end{array}$$

(56)$$\begin{array}{l} \Xi_{6}=\sum_{n_{s}=0}^{N_{s}-1}\sum_{m=0}^{n_{s}}\left(\begin{array}{c} n_{s}\\ m \end{array}\right)\frac{t_{1}^{n_{s}}{\left(\beta {\bar{\gamma }}_{SR}\right)}^{N_{d}}{\left(\left(1-\beta \right){\bar{\gamma }}_{RD}\right)}^{m}e^{-\;\;\frac{t_{1}}{\left(1-\rho \right)\beta \lambda {\bar{\gamma }}_{SR}}}\Gamma \left(N_{d}+m,\frac{u_{3}u}{\beta {\bar{\gamma }}_{SR}{\bar{\gamma }}_{RD}}\right)}{n_{s}!{\left(\left(1-\rho \right)\beta \lambda {\bar{\gamma }}_{SR}\right)}^{n_{s}-m}u_{3}^{N_{d}+m}}. \end{array}$$

Based on Equations (Equation 54)–(Equation 56), when λ→∞, the asymptotical performance of the EST in the MRT-MRC scheme can be calculated as
(57)$$\begin{array}{l} \lim_{\lambda \to \infty }\zeta^{MRT-MRC}=\frac{R_{s}}{2}\left[1-\sum_{n_{s}=0}^{N_{s}-1}\;\right.\left.\frac{\left(N_{d}+n_{s}+1\right)!\;\;{\left(\beta {\bar{\gamma }}_{SR}\right)}^{N_{d}}{\left(\left(1-\beta \right)t_{1}{\bar{\gamma }}_{RD}\right)}^{n_{s}}}{n_{s}!\left(N_{d}-1\right)!\;{\left(\left(1-\beta \right)t_{1}{\bar{\gamma }}_{RD}+\beta {\bar{\gamma }}_{SR}\right)}^{N_{d}+n_{s}+1}}\right]. \end{array}$$

Equation (Equation 57) denotes that the EST of the MRT-MRC scheme is irrelevant with λ when λ is sufficiently high.

## 4. Simulation and Discussion

The Monte Carlo simulation was utilized to verify the analytical expressions carried out in Section 4, and compare the performance of the four schemes. For each figure, $R_{0}=2\;\mathrm{bit}/s/\mathrm{Hz}$, $R_{s}=1\;\mathrm{bit}/s/\mathrm{Hz}$, η=0.5, ω=0.8 and ${\bar{\gamma }}_{SR}={\bar{\gamma }}_{RD}=0$. The black solid lines represent the numerical analyses in all figures. The dashed-lines are the asymptotic analysis results.

Figure 2 illustrates the SOP and COP versus the transmit SNR under the four schemes, where β and ρ were set as 0.5. As shown in Figure 2, the derived analytical curves excellently match with simulation results, which validate our derivation. Several observations can be obtained as follows: (1) The SOP of the four schemes increases continuously with increasing the transmit SNR, whereas the COP decreases. This is because, as the transmit SNR increases, the received SNR at *R* and *D* are improved. Therefore, there is a trade-off between security and reliability. (2) The SOP of each scheme approaches saturation when the transmit SNR is sufficient high. The reason is that the utilization of jamming signal protects the confidential information. (3) Both the TAS-RAS scheme and TAS-MRC scheme are superior to MRT-RAS scheme and MRT-MRC in terms of security performance, while inferior to MRT-RAS scheme and MRT-MRC in terms of reliability performance. It denotes that the security performance of TAS scheme is superior to MRT, while the reliability performance of RAS scheme is worse than MRC.

Figure 3 depicts that the EST versus the transmit SNR under the four schemes. It can be easily observed that the analytical approximations are close to the exact simulation results for all schemes. It is shown that, when λ→∞, the EST approaches to a constant for each scheme, which verifies the asymptotical optimality property shown in Equations (Equation 28), (Equation 40), (Equation 51) and (Equation 57). Therefore, by applying excessive transmit power to improve the EST is meaningless. In the high transmit SNR regime, several observations can be drawn as follows: (1) The EST performance of TAS-MRC scheme outperforms MRT-MRC scheme, and TAS-RAS scheme outperforms MRT-RAS scheme. This is because that the source node may leak more confidential information to the untrusted relay when applying the MRT. (2) The saturation values of TAS-RAS and MRT-MRC are very close. However, TAS-RAS is less complicated than MRT-MRC. (3) The EST of TAS-MRC scheme is more advantageous than MRT-RAS scheme.

Figure 4 depicts the curves of the ESTs versus β for four schemes. It is shown that: (1) There is an optimal β value that maximizes the EST. When β is less than the optimal value, the EST increases, and when β is greater than the optimal value, the EST decreases. This is because, when β is small, the source allocated power is small, thus the reliable link cannot be established between *S* and *D*, which leads to small EST. While β is large, the security of the transmitted information cannot be guaranteed, which also degrades EST. (2) For a given β, the EST of MRT-MRC scheme outperforms MRT-RAS scheme, and TAS-MRC scheme outperforms TAS-RAS scheme. This is because utilizing the MRC technique can obtain superior reliability to RAS. (3) In terms of EST, MRT-MRC scheme outperforms TAS-MRC scheme when $\beta \in \left(0,0.7\right)$, and then with continuously increasing β, MRT-MRC scheme is inferior to TAS-MRC scheme. Similarly, MRT-RAS scheme outperforms than TAS-RAS scheme when $\beta \in \left(0,0.675\right)$. This is explained as, with increase of β, the allocated power for the source increases, and the reliability of the system would improve. However, the allocated power for the destination decreases, and this leads to worse security performance.

Figure 5 depicts the EST with the number of antennas $N_{s}$ at the source node under four schemes. The figures show that: (1) The EST increases and then decreases when $N_{s}$ is increasing, thus there is an optimal $N_{s}$ that maximizes the EST. For MRT and TAS, the optimal $N_{s}$ is 4 and 17, respectively. This can be explained as increasing $N_{s}$ can improve the connection reliability. Therefore, the EST would increase firstly. However, the source node may leak more information to the untrusted relay with continuously increasing $N_{s}$, which degrades the security performance. (2) When the source antennas are more than 4, the EST of MRT-MRC scheme and MRT-RAS scheme deceases rapidly, whereas the EST of TAS-MRC scheme and TAS-RAS scheme declines slowly when the antenna number is higher than 17. This implies that MRT is more sensitive to the number of the source antennas compared with TAS.

Figure 6 shows that the EST versus the number of antennas $N_{d}$. It is observed that: (1) The EST increases with the increase of $N_{d}$ and then approaches a saturation value. This is because increase of $N_{d}$ improves the security of the system. The growing trend would saturate because the power of the destination is constrained. (2) The EST of MRT-MRC scheme outperforms TAS-MRC scheme, and MRT-RAS scheme outperforms TAS-RAS scheme. The reason is that MRT outperforms TAS for the source node. (3) The EST of MRT-MRC scheme outperforms MRT-RAS scheme, and TAS-MRC scheme outperforms TAS-RAS scheme. The reason is that MRC is more advantageous than RAS for the destination node.

Figure 7 depicts the curves of the derived EST versus ρ for four schemes. It is observed that: (1) There is an optimal ρ that maximizes the EST. Smaller ρ implies the harvested energy at *R* is smaller, which leads to the lower EST. For a larger ρ, the harvested energy for information transmission is small, which also degrades the EST. (2) The EST of MRT-MRC scheme is superior to TAS-MRC scheme, and MRT-RAS scheme outperforms TAS-RAS scheme. The EST of MRT-MRC scheme is superior to MRT-RAS scheme, and TAS-MRC scheme outperforms TAS-RAS scheme. The reason is as same as for the results shown in Figure 6. (3) The EST of TAS-MRC scheme is very close to MRT-RAS scheme.

Figure 8 shows the EST of the four schemes versus ω. Here, the ω proportion of the harvested energy is utilized for forwarding the confidential information to *D* and the other proportion is used for eavesdropping and decoding consumption. The results show that: (1) The EST is increasing according with the increase of ω. The reason is that, when ω increases, the allocated harvested energy at *R* for forwarding the confidential information to *D* is improved. As a result, the harvested energy for eavesdropping and decoding the information is reduced at *R*. Therefore, the security performance is promoted. (2) MRT-MRC scheme and MRT-RAS scheme are superior to TAS-MRC scheme and TAS-RAS scheme, respectively.

## 5. Conclusions

This paper presents secure transmission for a half-duplex AF untrusted and energy harvesting relay network under the PSR strategy, where both *S* and *D* are multi-antennas and *R* is a single antenna. The expressions of SOP, COP and EST for the four schemes are successively derived. Moreover, the EST asymptotic analysis of each scheme is provided in the high transmits SNR regime. These expressions were verified by Monte-Carlo simulation. The effects of various key parameters on the EST, such as λ, β, ρ, $N_{s}$ and $N_{d}$, were taken into account. In general, the EST of MRT-MRC scheme was superior to the other schemes in terms of the parameters β, $N_{d}$, ρ, and ω. However, the EST of MRT-MRC scheme was inferior to TAS-MRC scheme in the high transmit SNR regime. There existed an optimal $N_{s}$ to maximize the EST of the proposed schemes, i.e., 4 or 17 antennas.

## Figures and Tables

**Figure 1 sensors-19-00076-f001:**
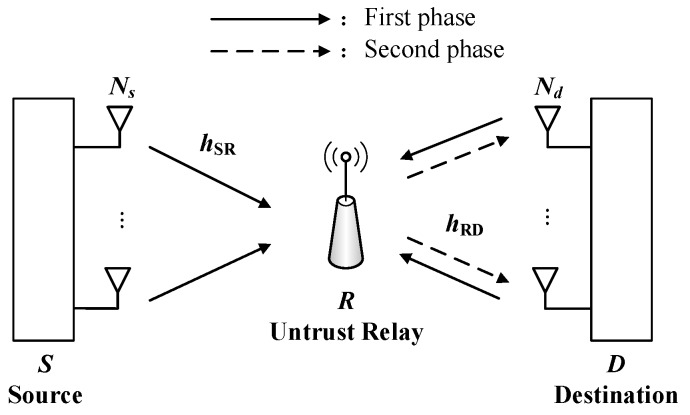
System model.

**Figure 2 sensors-19-00076-f002:**
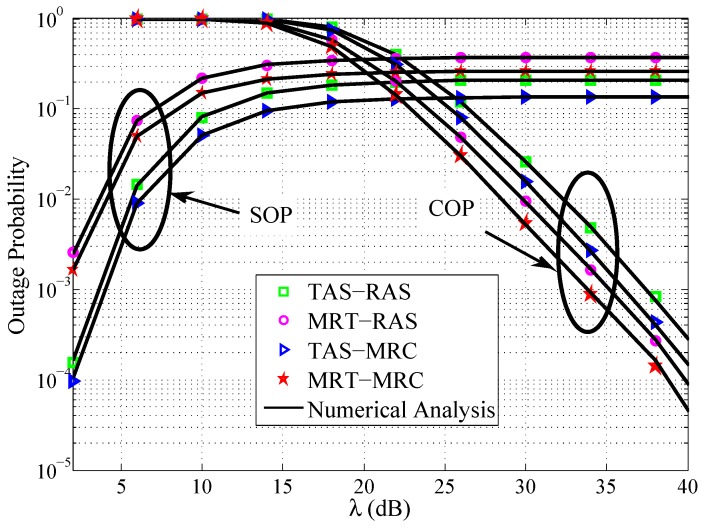
SOP and COP vs. the transmit SNR.

**Figure 3 sensors-19-00076-f003:**
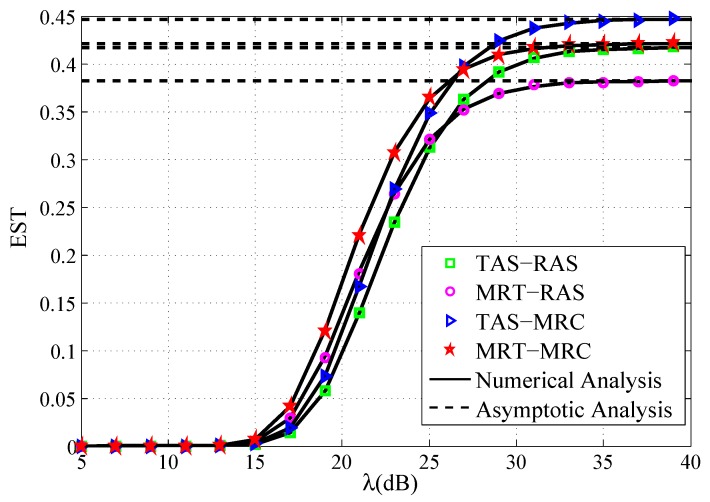
EST vs. the transmit SNR.

**Figure 4 sensors-19-00076-f004:**
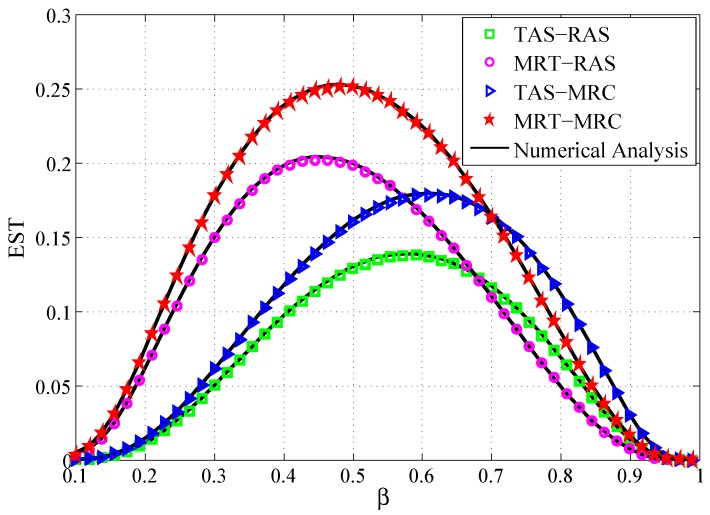
EST vs. the power allocation factor β.

**Figure 5 sensors-19-00076-f005:**
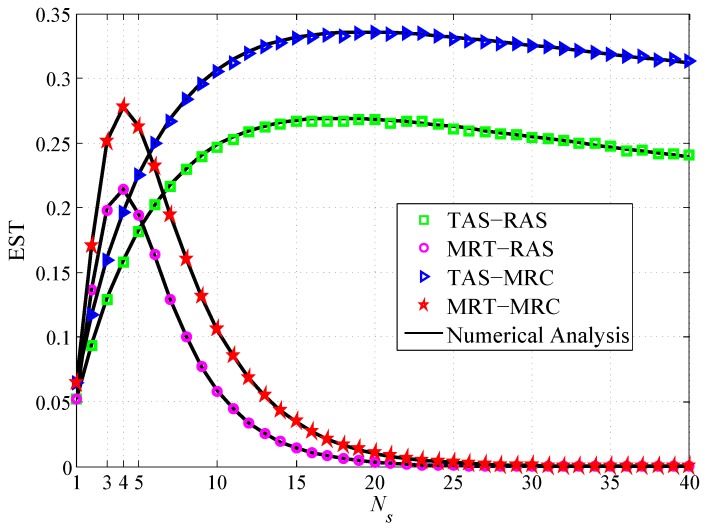
EST vs. the number of antenna $N_{s}$.

**Figure 6 sensors-19-00076-f006:**
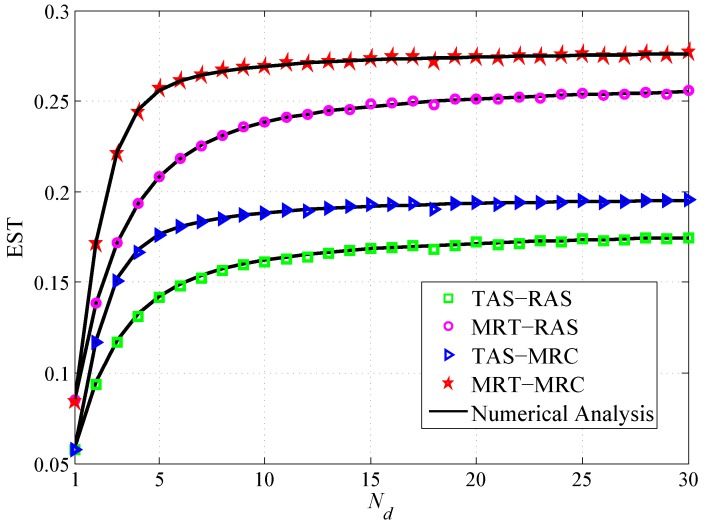
EST vs. the number of antenna $N_{d}$.

**Figure 7 sensors-19-00076-f007:**
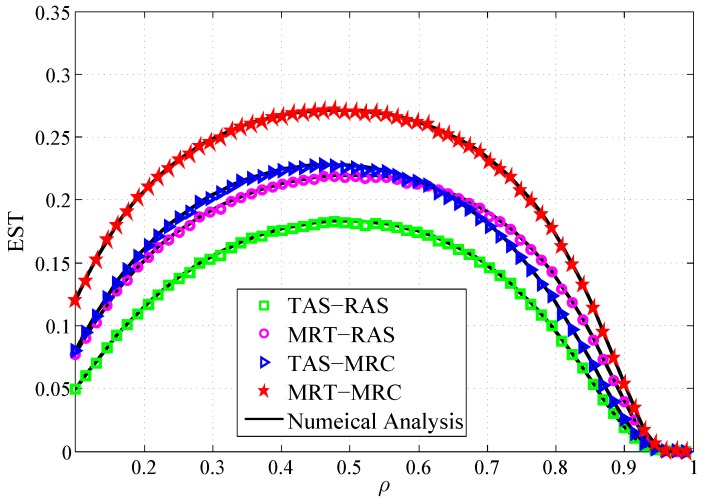
EST vs. the power splitting ratio ρ.

**Figure 8 sensors-19-00076-f008:**
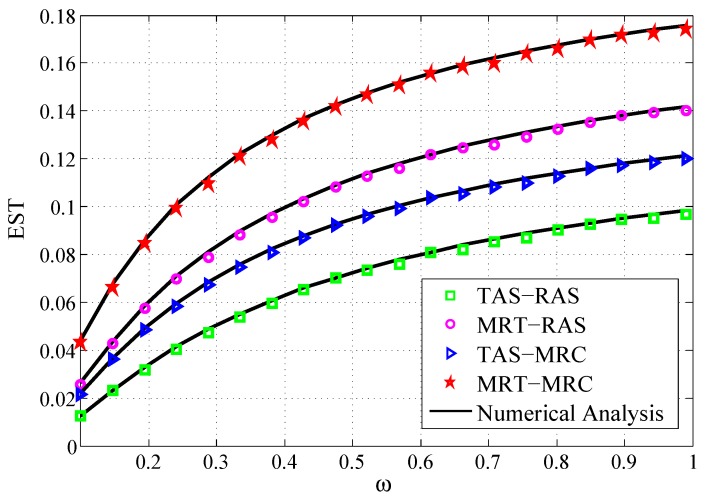
EST vs. the power splitting ratio ω.

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
