# Peer review of "Secure Transmission for Simultaneous Wireless Information and Power Transfer in AF Untrusted Relay Networks"

_sensors, 2018, doi:10.3390/s19010076_

Round 1

Reviewer 1 Report

This paper considers a cooperative AF relaying system with a multi-antenna source, multi-antenna destination, and a single-antenna relay that is assumed to be untrusted and empowered by energy harvesting. All involved wireless channels are assumed perfectly known and IID Rayleigh. The destination is assumed to send jamming signals during the first hop communication of the source's signal confidential to the untrusted relay. The authors consider different diversity precoding and combining schemes and present formuals for the system's secrecy performance. This reviewer has the following major concerns:

1) The proposed system configuration is not well motivated and its contribution compared with the state of the art seems marginal. Very similar setups with the same idealistic assumptions have appeared in available papers. So, the authors need to motivate their system and consider practical nonidealities. 

2) Energy harvesting requires short distances and its gains are still practically questioned. In this paper it is expected that the AF relay does both energy harvesting and eavesdropping. How feasible is this under a practical setting?

3) Another major issue of this paper is related to the various considered idealities and missing literature that could be used in making the considered system more realistic and smarter. For example, the jamming signal could be designed in a smart way (see eg [R1] and [R2]) and the considered channel should include shadowing etc (see eg [R3]). On another note, the multiple antennas could be used to assist energy harvesting (see eg [R4] and [R5]), or even consider cooperative jamming [R6].

[R1] Securing relay networks with artificial noise: An error performance based approach, MDPI Entropy 19 2017.

[R2] Cooperative Secret Communication with Artificial Noise in Symmetric Interference Channel, IEEE COML 2010.

[R3] Secrecy outage analysis over correlated composite Nakagami-m/Gamma fading channels, IEEE COML 2018.

[R4] Transmit precoding and receive power splitting for harvested power maximization in MIMO SWIPT systems, IEEE TGCN 2018.

[R5] Jointly optimal spatial channel assignment and power allocation for MIMO SWIPT systems, IEEE WCL 2018.

[R6] Secure communications with cooperative jamming: Optimal power allocation and secrecy outage analysis, IEEE TVT 2017.

Author Response

We thank Reviewer 1 for his/her constructive comments. We believe that we have addressed
all the concerns in the revised manuscript.

Reviewer 2 Report

This is a well written and well presented paper. The research study is new and timely in the area of physical layer security, energy harvesting and relay. The concept of untrusted relay, although not new, it has not been considered in the context as explained by the reviewer. The research gap is justified in the last paragraph of the introduction before the contributions. The paper presents very interesting results that show how SOP, COP and EST performance parameters vary across different schemes and SNR regimes. The paper also show relevant graphs when power, power allocation factor and number of antennas is varied for different techniques.The results are well justified and well commented on and the graphs makes sense. The conclusions that derive from the results are interesting and add to the research in this area, hence I have rated the paper high.

I only have some minor revisions:

In the introduction, two references are relevant in this area and would complete the manuscript. They are both in the context of physical layer security, with relay and energy harvesting. One take into account also another hop (the backhaul link) and one  with Nakagami fading. The references are:

"Secure energy harvesting relay networks with unreliable backhaul connections." IEEE Access 6 (2018): 12074-12084

 "Secure Energy Harvesting Communications with Relay Selection over Nakagami-m Fading Channels." Mobile Networks and Applications (2017): 1-8.

In the introduction, last paragraph before the contribution, it would be very useful to explain and justify why the trade-off between SOP and COP that has not been studied before is so important. Explain this and also explain why the parameter EST is important.

The parameter beta from fig.4 and ro from fig.7 are explained but the parameter omega that appears in equation 2 and in figure 8 is not explained. Explain

R that appears in the text before eq.13 stands for Relay, mention that, it is confusing when it is in the text with all the rates, which are also Re, Ro, Rs, etc

"Plugging" appears before eq. 23 and before eq. 48, use a more formal word, such as "substituting"?

The simulation parameters are just mentioned at the beginnning of section 4 but not justified.  why do you choose Ro=2bit/s/Hz, mu=0.5, omega=0.8. justify if possible with references. Are you thinking on any particular technology in your system? would the results vary significantly if different values of Ro are chosen for example?

Before fig. 3 you make an interesting observation, applying excessive tranmsit power to improve the EST is meaningless. Is this all across SNR regimes, only for low and high? Also this is very interesting, elaborate this point, what would be the consequences of applying excessive transmit power, specially if there is nothing to gain.

In fig. 4 I can see that there are optimum values of beta without reading the comments and I can see which techniques are better than others. You justify why some techniques are better well in your comments but do you have any justification on why the optimum point is displaced and not the same for different techniques (for example why MRT-RAS achieves optimum for lower beta values than all other schemes curves).

In fig.8 you evaluate EST against w, omega. All results in fig 8 are well justified but explain the importance of omega.

Author Response

We thank Reviewer 2 for his/her valuable comments that help to improve the quality of this
manuscript. We believe that we have addressed all the concerns in the revised manuscript.

Reviewer 3 Report

This paper studied secure transmission for a half-duplex AF untrusted and energy harvesting relay network under the PSR strategy, with both S and D being equipped with multiple antennas and R being a single-antenna device. The topics of energy harvesting and physical layer security are worthwhile and timely. However, within the context of available contributions on the areas, the novelty and technical contributions of the paper are questionable.

1) The system model is not sophisticated since a single-antenna relay was considered. A multiuser scenario would render the analysis more practical as well.

2) Some assumptions must be better justified. In particular, the consideration of perfect channel state information is rather unrealistic. The same applies to the consideration of non-feedback delay. I understand that such assumptions jeopardize the technical contribution and usefulness of the proposed analysis.

3) What is the reason for assuming PSR? How about time-switching relaying strategy? Isn't it more useful and practical?

4)  In terms of deriving the outage probability, what are the significant challenges solved? For example, since most of the results related to random variables and distributions are available, the derived results appear to be of incremental value. Please comment and explain if any novel result? 

5) Comparisons with other precoding and multi-antenna schemes could be carried out.

Author Response

We thank Reviewer 3 for his/her valuable comments that help to improve the quality of this
manuscript. We believe that we have addressed all the concerns in the revised manuscript.

Round 2

Reviewer 1 Report

The authors provided convincing answers to my concerns.

Reviewer 3 Report

The authors did not properly address my raised concerns. The paper still contains the same drawbacks of the previous submission, and due to this fact my decision now is "Reject". The details are given below.

1) Regarding my first comment related to the consideration of a multi-user scenario, the authors' reply was unsatisfactory. Actually, an attempt to consider a special case of a multi-user scenario was not even considered. Only a few papers on single-user systems were mentioned in the reply, and I understand this is not the right way to reply a raised concern. Note that the literature has already a great number of works dealing with the multi-user scenario, but the authors neglected such works along the discussions.

2) The assumption of imperfect CSI was also disregarded by the authors although it is essentially the most realistic issue. Again, the authors' reply moved the discussions to what was considered in the paper, and such a key point was neglected. Showing previous works on the adopted assumption while neglecting previous ones on the raised concern is definitely the wrong way to reply a specific question.

3) Comparisons with previous precoding schemes were not carried out as well. It is one more unaddressed concern.

In summary, the authors failed to reply and to address my comments. Owing to this, my recommendation is now "Reject".

Author Response

Please find authors' response in the attachment.
